# Pectoral Fin Anomalies in *tbx5a* Knockdown Zebrafish Embryos Related to the Cascade Effect of N-Cadherin and Extracellular Matrix Formation

**DOI:** 10.3390/jdb7030015

**Published:** 2019-07-12

**Authors:** Jenn-Kan Lu, Tzu-Chun Tsai, Hsinyu Lee, Kai Hsia, Chih-Hsun Lin, Jen-Her Lu

**Affiliations:** 1Laboratory of Molecular Biology, Institute of Aquaculture, National Taiwan Ocean University, Keelung 20224, Taiwan; 2Institutes of Clinical Medicine; Department of Surgery and Pediatrics, School of Medicine, National Yang-Ming University, Taipei 11221, Taiwan; 3Department of Pediatrics, Taoyuan General Hospital, Ministry of Health and Welfare, Taoyuan 33004, Taiwan; 4Department of Life Science, National Taiwan University, Taipei 10617, Taiwan; 5Department of Pediatrics, Taipei Veterans General Hospital, No. 201, Section 2, Shih-Pei Rd., Beitou, Taipei 11217, Taiwan; 6Department of Plastic and Reconstructive surgery, of Surgery, Taipei Veterans General Hospital, No. 201, Section 2, Shih-Pei Rd., Beitou, Taipei 11217, Taiwan

**Keywords:** chondrogenic condensation, *tbx5a*, N-cadherin, chondrogenesis, fibronectin, hyaluronic acid, endoskeleton formation

## Abstract

Functional knockdown of zebrafish *tbx5a* causes hypoplasia or aplasia of pectoral fins. This study aimed to assess developmental pectoral fin anomalies in *tbx5a* morpholino knockdown zebrafish embryos. The expression of cartilage-related genes in the *tbx5a* morphant was analyzed by DNA microarray, immunostaining, and thin-section histology to examine the detailed distribution of the extracellular matrix (ECM) during different pectoral fin developmental stages. Chondrogenic condensation (CC) in the *tbx5a* morpholino knockdown group was barely recognizable at 37 h postfertilization (hpf); the process from CC to endoskeleton formation was disrupted at 48 hpf, and the endoskeleton was only loosely formed at 72 hpf. Microarrays identified 18 downregulated genes in *tbx5a*-deficient embryos, including 2 fin morphogenesis-related (*cx43*, *bbs7*), 4 fin development-related (*hoxc8a*, *hhip*, *axin1*, *msxb*), and 12 cartilage development-related (*mmp14a*, *sec23b*, *tfap2a*, *slc35b2*, *dlx5a*, *dlx1a*, *tfap2b*, *fmr1*, *runx3*, *cdh2*, *lect1*, *acvr2a*, *mmp14b*) genes, at 24 and 30 hpf. The increase in apoptosis-related proteins (BAD and BCL2) in the *tbx5a* morphant influenced the cellular component of pectoral fins and resulted in chondrocyte reduction throughout the different CC phases. Furthermore, *tbx5a* knockdown interfered with ECM formation in pectoral fins, affecting glycosaminoglycans, fibronectin, hyaluronic acid (HA), and N-cadherin. Our results provide evidence that the pectoral fin phenotypic anomaly induced by *tbx5a* knockdown is related to disruption of the mesoderm and ECM, consequently interfering with mesoderm migration, CC, and subsequent endoskeleton formation.

## 1. Introduction

Chondrogenesis is one of the earliest events in the morphogenesis of the pectoral fin [1]. The genetic pathways of the pectoral fin in zebrafish and forelimbs in tetrapods are highly conserved [2]. Holt–Oram syndrome is an inherited disorder in humans characterized by upper limb deformity and congenital heart disease, and is related to mutation of the *tbx5a* (T-box transcription factor 5) gene [3,4,5,6]. However, the molecular and cellular processes underlying upper limb agenesis or hypoplasia in Holt–Oram syndrome remain unclear.

The *tbx5a* gene is expressed in the early stage of pectoral fin development in zebrafish embryos [7,8]. The bilateral protrusions of the mesoderm in the lateral plate are pectoral fin precursor cells that form the pectoral fin buds in zebrafish. The expression of the *tbx5a* gene in the lateral plate mesoderm at 10–15 h postfertilization (hpf) constitutes the earliest specific marker of pectoral fin development in zebrafish [9,10,11]. The signaling center of pectoral fin buds in zebrafish embryos is located in the apical ectodermal ridge (AER). Formation of the pectoral fin includes blastema formation, and pectoral fin bud formation in zebrafish embryos is initiated at 26 hpf [12]. Mesenchymal cells cluster to form oval-shaped pectoral fin buds at 32 hpf. The formation of the AER is initiated at 36 hpf via chondrogenic condensation (CC) [13].

The mesenchymal cell migration during fin formation, which plays a crucial role in the localization and timing of pectoral fin bud induction, depends on the expression of the *tbx5a* gene [14,15,16]. The *tbx5* gene is essential for initiating AER signaling in mesenchymal cells until the end of pectoral fin bud formation at 48 hpf. Furthermore, *tbx5a* expression is maintained at a low level up to 96 hpf [17,18,19]. Defects in the *tbx5a* gene result in variable scapulocoracoid, endoskeleton disc, hypoplasia, and distal cleithrum dislocation defects. These limb bud-derived abnormalities consequently cause agenesis, shortening, or deformity of the pectoral fin [9,18,20]. The functional knockdown of zebrafish *tbx5a* by morpholino (MO) administration results in a failure of mesodermal cell migration, which leads to the failure of initiating fin bud formation and suggests that *tbx5a* functions very early in the pectoral fin induction pathway [14].

This study aims to assess cellular and extracellular matrix (ECM) components in the pectoral fin of *tbx5a* gene knockdown zebrafish embryos. The expression of cartilage-related genes in *tbx5a* gene knockdown zebrafish embryos was studied by DNA microarrays and immunohistology to examine the distributions of proteoglycans (PGs) and neural cadherin (N-cadherin) at different pectoral fin developmental stages.

## 2. Materials and Methods

### 2.1. Animal Ethics

Approval for this experiment was granted by the Animal Ethics Review Board of National Taiwan Ocean University Aquaculture (IACUC 105031, College of Life Science). Since zebrafish embryos under 7 days (168 hpf) are excluded from the definition of “vertebrate animals” provided by the review board, our study, which used zebrafish embryos under 96 hpf, was spared the regulation and review process required by the Basic Institutional Review Board (IRB).

### 2.2. Maintenance and Breeding of Zebrafish and Embryo Collection

We used the Zebrafish AB strain carrying a specific transgene (Tg, cmlc2::EGFP; cmlc2:H2AFZmCherry)^cy3^ in this study. The details of the maintenance and breeding of zebrafish and embryo collection have been previously published [21,22,23].

### 2.3. Microinjections and Morpholino Treatment

Total RNA was prepared from defective or normal embryos in each group (Invitrogen Corporation, Carlsbad, CA, USA). The MO antisense oligonucleotide *tbx5a*-MO (5′-GAAAGGTGTCTTCACTGTCCGCCAT-3′) was designed against the *tbx5a* translational start site, and a mismatch-*tbx5a*-MO (5′-GTCTCTTGACTCTCCGCGATCTCGG-3′) was designed as a control (Gene Tools LLC, Philomath, OR, USA). Wild-type (WT) embryos, primarily at the one-cell stage with an intact chorion, were injected with 19.4 ng/4.3 nL stock MO diluted in Danieau’s buffer (*n* = 50 per group in triplicate). Three control groups, including (1) Control MO-injected group injected with the 3′ end of *tbx5a*-MO, (2) Blank microinjection group, and (3) Uninjected group (no microinjections), were included to identify the specific effects of blocking *tbx5a* mRNA translation with *tbx5a*-MO (*n* = 50 per group) (Appendix A).

### 2.4. Validation of the Efficacy, Traumatic Effects, and Off-Target Effects of tbx5a MO

To rule out the possibility that pectoral fin anomalies were induced by microinjection or a nonspecific off-target effect of *tbx5a* MO, we used early microinjection of missense *tbx5a* (Control MO-injected group) into the 1–4 cell stage of WT embryos as a control. We used Western blot to validate the efficacy of *tbx5a* MO. The off-target effect of microinjected *tbx5a* MO at the 1–4 cell stage was further evaluated by longitudinal follow-up studies. We used several indexes, including the longitudinal survival rate and heart rate follow-up, heart defect rate, and rate of trunk deformity (Appendix A).

### 2.5. Histology

At 48 hpf, the embryos were fixed in 4% paraformaldehyde, dehydrated using an ethanol series, cleared in xylene, and embedded in paraffin wax. Longitudinal sections were cut, dewaxed in xylene, and stained with hematoxylin and eosin (HE).

### 2.6. Alcian Blue Staining

Formaldehyde-fixed and paraffin-embedded tissue sections were processed in 70% ethanol for 3 min at room temperature and then incubated in 3% acetic acid for 3 min. The tissues were immersed in Alcian blue (ScienCell Research Lab, Carlsbad, CA, USA) solution (1% Alcian blue in 3% acetic acid) for 30 min. After tissue sections were washed in running tap water for 1 min and rinsed in diH_2_O for 2 min, the sections were dehydrated in two changes of 95% ethanol for 2 min each and cleared in xylene substitute (ScienCell Research Lab, CA, USA).

### 2.7. Immunohistochemistry Assay

At 30 hpf, embryos were fixed with 4% paraformaldehyde in phosphate-buffered saline (PBS). The deparaffinized sections (3 µm) were incubated with target-purified rabbit primary antibodies against HA (AnaSpec Inc., Fremont, CA, USA) and fibronectin (AnaSpec Inc., Fremont, CA, USA), washed with PBS, and incubated with rhodamine-conjugated goat anti-rabbit immunoglobulin G (IgG). Control samples were processed in parallel, omitting the primary antibody. Confocal images were obtained using a Zeiss LSM 880 confocal microscope with a 20× objective. Zen image acquisition software (Carl Zeiss, Jena, Germany) was used to analyze images. Samples of the *tbx5* morphant group at specific stages of embryo development were detected by the argon laser at wavelengths of 488 nm and 530 nm with a 20× objective lens and compared to those of the Uninjected and Control MO injection groups. Samples were scanned by controlling the laser wavelength between 510~540 nm and 580~650 nm to yield green and red fluorescence, respectively. The optimal aperture (pinhole) was adjusted by the receiver range, noise ratio control of the detector (amplification offset and amplification gain), and laser beam intensity. The best fluorescence intensity balance of the tissue image capture resolution was set to 1024 × 1024 to minimize the impact of background fluorescence. The images were reconstructed and observed using LCS Lite software (version 2.0) (Leica, Bannockburn, IL, USA).

### 2.8. RNA Isolation

Total RNA was isolated from 50 embryos using a guanidine isothiocyanate-based TRIzol solution. RNA samples were resuspended in diethyl pyrocarbonate (DEPC)-treated water and quantified spectrophotometrically at 260 nm. RNA quality was checked by 1.2% agarose gel electrophoresis after staining with 1 µg/mL ethidium bromide. The RNA stock solution was stored at −80 °C.

### 2.9. Microarray Analysis

Microarray analysis (WT vs. MO) was performed to determine the effect of *tbx5* deficiency. RNA from the WT and MO groups was isolated at 24, 30, and 48 hpf and purified using a RNeasy^®^ Mini Kit (QIAGEN, Hilden, Germany). RNA quality was confirmed using an Agilent 2100 Bioanalyzer (Agilent Technologies, Santa Cruz, CA, USA). Purified RNA was reverse-transcribed into cDNA using SuperScript TM III RT (Invitrogen, Carlsbad, CA, USA). Before purifying and coupling with a fluorescent dye by indirect cDNA labeling using a zebrafish-specific microarray kit covering 43,663 gene transcripts (Invitrogen, Carlsbad, CA, USA), cDNA was hydrolyzed and neutralized using NaOH and HCl. Then, the cDNA was pretreated with the GEx hybridization buffer HI-PRM (Agilent Technologies, Santa Cruz, CA, USA) before being transferred to hybridization chamber gasket slides for the hybridization reaction. The slides were scanned using a GenePix 4000B Axon Instruments scanner (Molecular Devices, Silicon Valley, CA, USA), and the data were analyzed using Genespring GX 10.0.2 (Agilent Technologies, Santa Cruz, CA, USA). All gene transcripts (43,663) were screened at these three time points. All data were compliant with the Minimum Information about a Microarray Experiment guidelines, and the raw data were deposited into the Gene Expression Omnibus (GSE33965, NCBI tracking system #16217606) as previously described [17,18]. Microarray data were analyzed based on gene expression data, expression ratios and normalized intensities. Housekeeping genes (w.r.t GAPDH) were used as a reference for normalization, and all other conditions were normalized with respect to the reference to obtain expression ratios. Each gene expression value in a single array experiment was divided by the mean expression values of these housekeeping genes. The expression of a certain gene in the Uninjected group was used as a reference and set to a value of 1. If the expression level of the same gene in the *tbx5a* morphant group was more than 1.5-fold higher than that in the Uninjected group, the gene was marked as “upregulated”. Conversely, if the expression level of the same gene in the *tbx5a* morphant group were more than 1.5-fold lower than that in the Uninjected group, the gene was marked as “downregulated”. These comparisons were analyzed at 24, 30, and 48 hpf. Pathway software was used to analyze fin morphogenesis and chondrogenesis. The expression level of each candidate gene that exceeded or fell below the threshold was further verified by quantitative RT-PCR at different developmental stages.

### 2.10. Quantitative Reverse-Transcriptase Polymerase Chain Reaction (qPCR)

Total RNA from 50 defective or normal embryos was prepared by the amplification of 3 µL of first-strand cDNA (Invitrogen). The amplification primer for the *cdh2* mRNA was obtained from published sequences as follows: *cdh2* (F: 5′-TGG CAA GAG GAC AAG GCG AGG ACG A-3′; R: 5′- GTG GGC AAT CAC TGG GTT GGG GCA-3′). The PCR conditions included denaturation at 95 °C for 3 min followed by 50 cycles of amplification (95 °C for 20 s, 59 °C for 15 s, and 72 °C for 20 s).

### 2.11. Statistical Analysis

All experiments were repeated at least three times, and the data are shown as the mean ± SD. The paired sample *T* test was applied for statistical analysis between experimental groups, and *p* < 0.05 was considered significant. One-way ANOVA with Duncan’s post hoc multiple range test was used to compare data among more than two groups, and *p* < 0.05 was considered significant. Statistical analysis was performed using IBM SPSS Statistics 19 (version 19; SPSS, Chicago, IL, USA).

## 3. Results

### 3.1. Abnormal Phenotypes of Pectoral Fins

After microinjection of *tbx5a* MO, embryos showed developmental defects in the pectoral fin at 96 hpf. The pectoral fin in the *tbx5a* morphant group showed unilateral or bilateral hypoplasia with a shortened length (Figure 1C–E). These abnormal hypoplastic limb buds exhibited outward flexion and/or were flat in shape (Figure 1A). The development of the limb bud was disrupted in most members of the *tbx5a* morphant group. In the *tbx5a* morphant group, 12% of the embryos showed normal fin development (6/50) at 48 hpf, 10% (5/50) showed normal fin development at 72 hpf, and 8% (4/50) showed normal fin development at 96 hpf (*n* = 50 each, triplet) (Figure 1F).

To rule out the possibility that pectoral fin anomalies were induced by microinjection or a nonspecific off-target effect of *tbx5a* MO, we performed early microinjection of missense *tbx5a* (Control MO-injected group) into WT embryos at the 1–4 cell stage as a control. All of these embryos showed normal pectoral fins without anomalies (50/50, 100%, *n* = 50 each, triplet) (Figure 1F). This result suggested that the possibility of an off-target effect of *tbx5a* MO inducing anomalies during fin development in this study is negligible.

### 3.2. Depressed tbx5a Gene Expression

At 48 hpf, *tbx5a* expression was restricted to the pectoral fin buds, eyes, and heart (Figure 2A,C). At 48 hpf, high levels of *tbx5a* expression were detected throughout the fin buds. Expression of *tbx5a* in the distal part of the pectoral fin buds was more intense than that in the proximal region (Figure 2A,C). Moreover, *tbx5a* expression in pectoral fins was significantly reduced in the *tbx5a* morphant at 48 hpf (Figure 2B,D).

### 3.3. Microarray Screening

We used a zebrafish chip (DNA microarray) to screen pectoral fin development-related gene transcripts. The total number of screened gene transcripts was 43,663, and the cutoff point was a 1.5-fold change in expression. After knockdown of *tbx5a*, 2250 gene transcripts were upregulated at 24 hpf, 2493 were upregulated at 30 hpf, and 2601 were upregulated at 48 hpf. After knockdown of *tbx5a*, 1248 gene transcripts were downregulated at 24 hpf, 747 were downregulated at 30 hpf, and 1468 were downregulated at 48 hpf.

Microarrays identified 18 downregulated gene transcripts related to fin development in the tbx5 morphant group at 24 and 30 hpf, but none of the gene transcripts related to cartilage development were upregulated in the tbx5 morphant group. The 18 downregulated gene transcripts comprised the following: 3 fin morphogenesis-related gene transcripts—*cx43* (connexin 43, which is related to transmembrane transport activity and gap junction alpha-1 protein), *bbs7* (Bardet–Biedl syndrome 7 is a protein-coding gene, related pathways are organelle biogenesis and maintenance and cargo trafficking to the periciliary membrane), and *hoxc8a* (homeobox C8a is a DNA-binding gene related to embryonic pectoral fin morphogenesis); 3 fin development-related gene transcripts—*hhip* (hedgehog interacting protein encoding gene, an important morphogen for anteroposterior patterns of limbs), *axin 1* (encodes a cytoplasmic protein that contains a G-protein signaling regulation domain and a disheveled and axin domain), and *msxb* (the encoded protein of muscle segment homeobox B is a transcriptional repressor whose normal activity may establish a balance between the survival and apoptosis of neural crest-derived cells required for proper craniofacial morphogenesis); and 12 cartilage development-related gene transcripts—*mmp14a* (the protein encoded by matrix metalloproteinase 14 alpha is involved in the breakdown of ECM in embryonic development and tissue remodeling), *sec23b* (Sec23 homolog B is an essential component of coat protein complex II-coated vesicles that transport secretory proteins from the endoplasmic reticulum to the Golgi complex), *tfap2a* (transcription factor AP-2 alpha is a protein-coding gene, and defects in this gene are a cause of branchio-oculo-facial syndrome), *slc35b2* (solute carrier family 35; member B2 is a protein-coding gene, and the related pathways are metabolism and glycosaminoglycan metabolism), *dlx5a* (distal-less homeobox gene 5a is related to embryonic viscerocranium morphogenesis and split-hand/foot malformation 1 with sensorineural hearing loss), *dlx1a* (distal-less homeobox gene 1a is related to the pharyngeal arch cartilage that forms the largest skeletal element of the ventral branchial arches), *tfap2b* (transcription factor AP-2 beta is related to the stimulation of cell proliferation and suppression of the terminal differentiation of specific cell types during embryonic development), *fmr1* (fragile X mental retardation 1 is related to multifunctional polyribosome-associated RNA-binding protein, which plays a central role in neuronal development and synaptic plasticity), *runx3* (runt-related transcription factor 3 is a protein-coding gene and related to cleidocranial dysplasia), *cdh2* (cadherin 2, a neuronal gene encoding a classical cadherin and a member of the cadherin superfamily, plays a role in the formation of cartilage and bone), *lect1* (leukocyte cell-derived chemotaxin 1 is expressed in the avascular zone of prehypertrophic cartilage, and its expression decreases during chondrocyte hypertrophy and vascular invasion), *acvr2a* (activin receptor IIa encodes a receptor that mediates the functions of activins, which are members of the transforming growth factor-beta (TGF-beta) superfamily involved in diverse biological processes), and *mmp14b* (matrix metalloproteinase 14 (membrane-inserted) beta MMP14 (matrix metalloproteinase 14), an essential protein-coding gene for pericellular collagenolysis and the modeling of skeletal and extraskeletal connective tissues during development) (Table 1).

### 3.4. Chondrocytes and Glycosaminoglycans in Pectoral Fins

HE and Alcian blue staining were used in histological sections of pectoral fins to reveal chondrocytes (Figure 3) and glycosaminoglycans (GAGs) in the cartilage (Figure 4). The apical thickening, apical fold, and beginning of the central CC in the pectoral fin could be observed in WT embryos at 37 hpf (Figure 3A). Further conversion of the CC to a tubular shape surrounded by ventral and dorsal myogenic mesenchyme was observed at 48 hpf (Figure 3C), and the formation of an endoskeletal disc with dorsal and ventral musculature was observed at 72 hpf (Figure 3E). The chondrocyte number was reduced (Figure 3B), and CC was severely depressed in the *tbx5a* morphant group at 37 hpf (Figure 3D). The formation of tubular-shaped cartilage tissue was barely visible at 48 hpf (Figure 3F).

Compared with the Uninjected group (Figure 4A,C,E), the tbx5 morphant group showed a reduction in GAGs in the cartilage (Figure 4B,D,F). High levels of GAGs were evenly distributed throughout the whole pectoral fin in WT embryos in the early stage of CC at 37 hpf (Figure 4A). By contrast, GAGs were sparse in the tbx5 morphant group (Figure 4B). Synchronized with the late phase of CC at 48 hpf, abundant GAGs were located in the central region of pectoral fins in the Uninjected group (Figure 4C). By contrast, GAGs were formed only loosely in the tbx5 morphant group at 48 hpf (Figure 4D). At the stage of endoskeleton formation, high levels of GAGs were concentrated centrally in the fin in the Uninjected group at 72 hpf (Figure 4E). The stunted pectoral fins in the *tbx5a* morphant group appeared substantially thickened with variable shortening of the proximal–distal length (Figure 4F). The endoskeletal disc in stunted pectoral fins showed multiple layers of disorganized chondrocytes with hypoplastic muscular development (Figure 3F and Figure 4F). The embryo with morphological bilateral hypoplasia/agenesis of the pectoral fin showed an absence of chondrocytes and GAGs at the aplasia site, disorganized GAGs in the hypoplastic pectoral fin (Figure 1G,H), and well-organized GAGs in the normal pectoral fin (Figure 1H,I).

### 3.5. Reduced cdh2 Gene Expression

After microinjection of 19.4 ng/2.3 nL *tbx5a* MO into zebrafish embryos at the 1–4 cell stage, embryos were collected at 26, 30, 37, and 48 hpf (*n* = 50, triplet). Compared with the Uninjected group, the *tbx5a* morphant group showed a significant reduction in *cdh2* expression at 26, 30, 37, and 48 hpf as determined by RT-qPCR (Figure 5G).

### 3.6. Reduction in N-Cadherin Protein Expression in Pectoral Fins

We used immunostaining to assess the localization of the N-cadherin protein (CDH2) in thin sections of pectoral fins. The formation of N-cadherin at 37 hpf was severely affected in the *tbx5a* morphant group compared with that in the control group, resulting in a severely hypoplastic pectoral fin (Figure 5). At the CC stage, the level of N-cadherin in the Uninjected group extended homogenously throughout the pectoral fin tissue at 37 hpf (Figure 5A), but only sparse N-cadherin was found in the *tbx5a* morphant group (Figure 5B). The levels of N-cadherin synchronized with the development of the dorsal and ventral myogenic mesenchymal mesoderm in the Uninjected group and extended homogenously throughout the pectoral fin tissue at 48 hpf (Figure 5C). In the *tbx5a* morphant group, the formation of N-cadherin was depressed and restricted in only the basement part of the pectoral fin at 48 hpf (Figure 5D). At the endoskeleton formation stage at 72 hpf, N-cadherin extended homogenously from the dorsal muscular structure to the trunk in the Uninjected group (Figure 5E). Both the ventral and dorsal muscular structures surrounding the endoskeleton myogenic mesenchyme exhibited high levels of N-cadherin in the Uninjected group (Figure 5F). The centrally located cartilage lacked N-cadherin and was well-formed at 72 hpf in the Uninjected group. Nevertheless, the perichondrial cells surrounding the forming cartilage in the Uninjected group exhibited high levels of N-cadherin (Figure 5E). The weak expression of N-cadherin could be detected in the stunted fins in the *tbx5a* morphant group at 72 hpf (Figure 5F).

### 3.7. Reduction in ECM Protein and Cx4 Expression

HA-binding protein and fibronectin in the pectoral fin were examined by immunostaining at 48 hpf. HA-binding protein (Figure 6A) and fibronectin (Figure 6C) were strongly expressed throughout the pectoral fin in the Uninjected group, but their expression levels were significantly decreased in the *tbx5a* morphant group (Figure 6B,D). Using real-time RT-PCR, we confirmed that *Cx43* was significantly downregulated in the *tbx5a* morphant group throughout different stages of pectoral fin formation (Figure 6E).

### 3.8. Increase in Apoptosis-Related Proteins in Pectoral Fins

The terminal deoxynucleotidyl transferase dUTP nick end labeling (TUNEL) assay demonstrated only a few apoptotic cells in the Uninjected group at 37 hpf (Figure 2E,G,I). However, a marked increase in apoptotic cells was observed in the head, heart, fin, and spine in the *tbx5a* morphant group (Figure 2F,H,J).

Two apoptosis-related proteins (B cell lymphoma 2 (BCL2) and BCL2-associated death promoter (BAD)) were examined by immunostaining at 30 hpf. Our results showed a remarkable increase in BCL2 and BAD protein expression in the *tbx5a* morphant group (Figure 7B,D) compared with that in the Uninjected group (Figure 7A,C). Using real-time RT-PCR, we confirmed that BAD (Figure 7E) and BCL2 (Figure 7F) were significantly upregulated in the *tbx5a* morphant group throughout the different stages of pectoral fin formation.

## 4. Discussion

Cartilage formation in developing vertebrate embryonic limbs consists of a series of events involving mesenchymal cell recruitment, migration, proliferation, and condensation [24]. CC is the critical stage in the development of cartilage and other mesenchymal tissues [1].

Genetic mutations, such as brachypod and phocomelia in mice and talpid in chickens, cause abnormal chondrogenesis during the CC stage and skeletal defects [25,26]. In this study, *tbx5a* knockdown resulted in early-onset alterations in chondrogenesis from mesenchymal cell migration onward. One of the earliest events in chondrogenesis in the pre-CC stage is the aggregation of chondroprogenitor mesenchymal cells. The microarray screening results in this report provided strong evidence that several genes involved in chondrogenesis were downregulated in the *tbx5a* morphant group at 24 and 30 hpf. A high cell density of chondroprogenitor mesenchymal cells is required for the occurrence of chondrogenesis, and the extent of cellular condensation correlates with the level of chondrogenesis [27]. In our study, the mesenchymal cell density of pectoral fins in larvae at the CC stage was high in the Uninjected group and severely reduced in the *tbx5a* morphant group. As demonstrated by our previous study, *tbx5a* deficiency provokes the expression of apoptosis-related genes distributed across multiple organs, reducing the promising maturation of *tbx5a* morphant embryos [28]. Using TUNEL and immunostaining assays, increases in apoptosis (BAD and BCL2)-related proteins were observed in pectoral fins in the *tbx5a* morphant group at the CC stage (30 and 37 hpf). Our findings suggested that mesenchymal cell reduction in the *tbx5a* morphant was related to cell apoptosis, which subsequently interfered with the morphogenesis of the pectoral fin and contributed to its hypoplasia or even agenesis in the *tbx5a* morphant group.

The cartilage ECM of zebrafish larvae is a polymorphic structure consisting of collagen I, collagen II, and large networks of PGs that contain GAGs, HA, fiber, and other molecular components, including fibronectin and laminin [29,30].

GAGs play an indispensable role during chondrogenesis to create highly organized elements in pectoral fin formation. GAGs are important protein carrier molecules in mesenchymal cell condensation, where they are involved in a wide range of signaling processes during pectoral fin development [31]. We used Alcian blue to stain GAGs in cartilage in transverse sections of pectoral fins and observed high expression levels in all stages in the Uninjected group. However, the formation of GAGs in the *tbx5a* morphant group was severely reduced at the CC and endoskeleton formation stages. The endoskeletal disc was well organized in the WT group at 72 hpf, but it was hypoplastic and disorganized with residual GAGs in the *tbx5a* morphant group.

We identified two GAG-related genes (*slc35B2* and *cdh2*) that were downregulated in the *tbx5a* morphant group via microarray analysis. We report, here, for the first time that *tbx5* gene expression affects *cdh2* gene expression and N-cadherin formation. In zebrafish, N-cadherin, encoded by the gene *cdh2*, is expressed extensively from late blastula stages onward [13]. *cdh2* function is crucial for the normal chondrogenesis of zebrafish pectoral limb buds by mediating functional mesenchymal cell condensation [32]. CC in the endoskeleton in *cdh2* knockdown embryos is missing or barely recognizable in the zebrafish fin bud [13]. These changes in the zebrafish limb bud in *cdh2* knockdown embryos are very similar to those in the *tbx5a* morphant group.

N-cadherin is a key cell adhesion molecule for chondroprogenitor mesenchymal cell–cell adhesion at the CC stage [33,34,35]. A lack of N-cadherin formation results in the failure of mesenchymal cell condensation, thus inhibiting subsequent chondrogenesis [36,37,38,39]. In this report, N-cadherin was normally present throughout the morphogenesis of the pectoral fin in the mesoderm-derived part of the paired larvae pectoral fins. High levels of N-cadherin were observed mainly at the basement of the pectoral fin bud and central trunk in the Uninjected group at the early stage of CC (37 hpf). These high levels of N-cadherin extended throughout the entire fin in the late CC stage (48 hpf) and were diminished at the endoskeleton formation stage (72 hpf). Nevertheless, the formation of N-cadherin in the zebrafish endoskeleton in the *tbx5a* morphant group was severely reduced throughout the entire CC stage. N-cadherin was completely absent in cartilage tissue at 72 hpf in the Uninjected group, whereas the perichondria cells surrounding the formed cartilage still exhibited high N-cadherin levels. The formation of N-cadherin surrounding the formed cartilage in the *tbx5a* morphant group was severely reduced at 72 hpf.

The microarray screening results provided strong evidence that multiple genes related to ECM formation were downregulated in the *tbx5a* morphant group at the pre-CC (24 hpf) and early CC stages (30 hpf). Fibronectin is a large ECM glycoprotein that affects N-cadherin cell–cell and cell–matrix adhesions [40]. The pectoral fin disk cartilage of zebrafish is notable for its fibronectin-rich ECM [41]. We examined fibronectin in the pectoral fin at 48 hpf and found abundant fibronectin in the Uninjected group, whereas fibronectin was significantly reduced in the *tbx5a* morphant group. Our results suggested that fibronectin was strongly upregulated at the condensation and differentiation stages. Depletion of TBX5 leads to abnormal fibronectin and fibrillin formation in xenopus embryos due to alterations in cardiac cell cycle progression [42]. We previously observed a similar relationship between abnormal fibronectin and cardiac cell cycle disruption at the stage of myocardium formation in *tbx5a* knockdown zebrafish [22,28].

*Cx43* regulates cell proliferation, fin size, and fin shape during the process of chondrogenesis in zebrafish embryos [43,44]. HA serves as a central organizing ECM molecule for binding to PGs and is molecularly and functionally downstream of *Cx43* during the process of cartilage morphogenesis [43]. Using DNA microarray screening and qPCR quantification, we report for the first time that *Cx43* expression was significantly inhibited in the *tbx5a* morphant group at 24, 30, 36, and 48 hpf. We examined HA in the pectoral fin at 48 hpf and found abundant HA in the Uninjected group, whereas HA was significantly reduced in the *tbx5a* morphant group.

Our findings support the conclusion that *tbx5a* deficiency causes both cellular and ECM disturbances during cartilage formation in the developing zebrafish pectoral fins. *tbx5a* deficiency induced the downregulation of several fin development-related genes. *tbx5a* knockdown likely results in reduced expression of FGF8, FGF10, and WNT signaling [15,17] which, in turn, results in reduced proliferation and migration of the mesoderm. The loss of *tbx5a* reduces the mesodermal component of the fin buds, AER formation and signaling, increased apoptosis, and/or decreased immigration of mesenchymal cells from the lateral plate. The disruption of ECM formation, including fibronectin, HA, GAGs, and N-cadherin, was identified in the pre-CC stage onward.

In conclusion, the phenotypic anomalies of the pectoral fin in the *tbx5a* morphant group included a disruption of mesodermal cell migration and CC and interference with subsequent endoskeleton formation, which was related to the *tbx5a* downregulation of a complex network of genes related to cell–cell communication and cell–matrix interactions. Thus, our results provide a context to understand the molecular and cellular processes of upper limb defects in Holt–Oram syndrome.

## Figures and Tables

**Figure 1 jdb-07-00015-f001:**
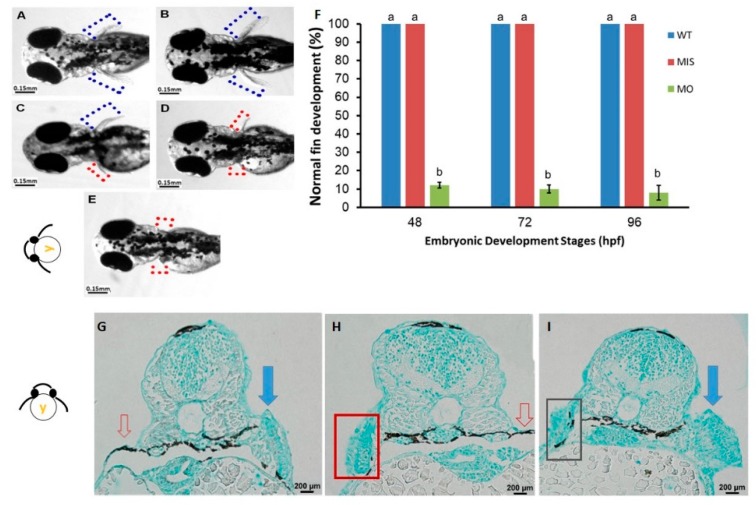
Phenotypes of pectoral fins in *tbx5a*-deficient embryos. Wild-type embryos with bilateral normal development of the pectoral fin (**A**,**B**). Unilateral hypoplasia (**C**), unilateral agenesis (**D**), and bilateral agenesis (**E**) of the pectoral fin in *tbx5a* morphants at 96 hpf. The rate of normal fin development was significantly reduced in *tbx5a*-deficient embryos at 48, 72, and 96 hpf (**F**). a, b: A significant difference was detected by one-way ANOVA with Duncan’s multiple range test. WT: Uninjected group, MIS: Control MO-injected group, MO: *tbx5a* morphant group. (**G**) Cross section of embryos with morphologic unilateral agenesis of the pectoral fin with Alcian blue staining showing well-organized glycosaminoglycans (GAGs) in the normal pectoral fin (blue arrow) with complete absence of chondrocytes and GAGs at the aplasia site (red arrow). (**H**) Cross section of the embryo with morphologic bilateral hypoplasia/agenesis of the pectoral fin showing the absence of chondrocytes and GAGs at the aplasia site (red arrow) with disorganized GAGs in the hypoplastic pectoral fin (red box). (**I**) Cross section of the embryo with morphologic unilateral hypoplasia of the pectoral fin showing disorganized GAGs in the hypoplastic pectoral fin (black box) and well-organized GAGs in the normal pectoral fin (blue arrow).

**Figure 2 jdb-07-00015-f002:**
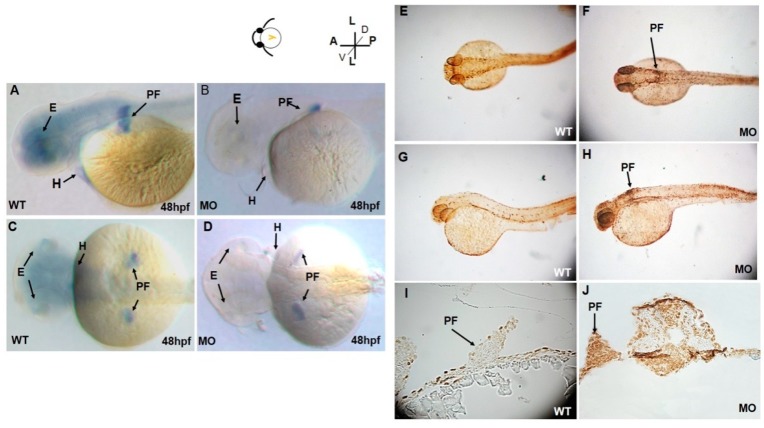
Lateral (**A**) and dorsal (**C**) views of *tbx5a* gene expression in the eyes, heart, and bilateral pectoral fins of wild-type embryos at 48 hpf. *tbx5a* gene expression in the eyes, heart, and pectoral fins was severely inhibited at 48 hpf in both the lateral (**B**) and ventral (**D**) views. Detection of apoptotic cells by the TUNEL assay in wild-type (**E**,**G**,**I**) and *tbx5a* knockdown (**F**,**H**,**J**) zebrafish embryos at 37 hpf. No TUNEL-positive cells are observable in the dorsal view (**E**), lateral view (**G**), or cross sections at the pectoral fin level (**I**) in wild-type embryos. However, massive numbers of TUNEL-positive cells in the pectoral fin region (black arrow) in the *tbx5a* morphant embryos were visible in the dorsal view (**F**), lateral view (**H**) and cross section at the pectoral fin level (**J**). WT, Uninjected group; MO, *tbx5a* morphant group; PF, pectoral fin; E, eye, H, heart.

**Figure 3 jdb-07-00015-f003:**
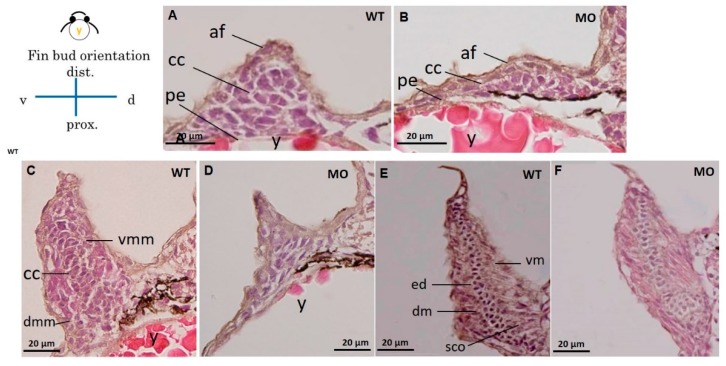
Cross section of a pectoral fin stained with HE. Wild-type embryos with well-developed myocytes and organized chondrocytes in pectoral fins at 37 (**A**), 48 (**C**), and 72 hpf (**E**). The hypoplastic pectoral fin of the *tbx5a* morphant showed reduced myocytes and disorganized chondrocytes at 37 (**B**), 48 (**D**) and 72 hpf (**F**). (af, apical fold; cc, chondrogenic condensation; dm, dorsal musculature; dmm, dorsal myogenic mesenchyme; ed. endoskeleton disc; vm, ventral musculature; vmm, ventral myogenic mesenchyme; pe, peritoneal epithelium; y, yolk, WT, Uninjected group, MO, tbx5 morphant group).

**Figure 4 jdb-07-00015-f004:**
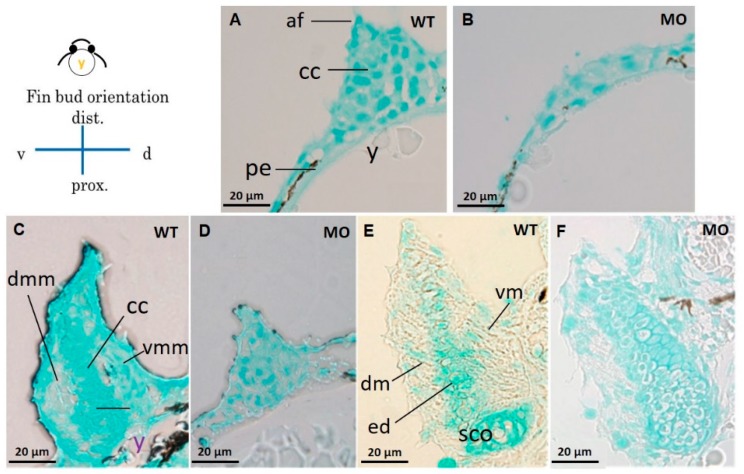
Cross section of pectoral fins stained with Alcian blue. Wild-type embryos showed well-developed apical folds and organized chondrocytes at 37 (**A**), 48 (**C**), and 72 hpf (**E**). The *tbx5a* morphant showed severe pectoral fin hypoplasia with loss of the apical fold, reduced chondrocytes, unorganized chondrogenic condensation and endoskeletal disc at 37 (**B**), 48 (**D**), and 72 hpf (**F**). (af, apical fold; cc, chondrogenic condensation; dm, dorsal musculature; dmm, dorsal myogenic mesenchyme; ed. endoskeleton disc; vm, ventral musculature; vmm, ventral myogenic mesenchyme; pe, peritoneal epithelium; y, yolk).

**Figure 5 jdb-07-00015-f005:**
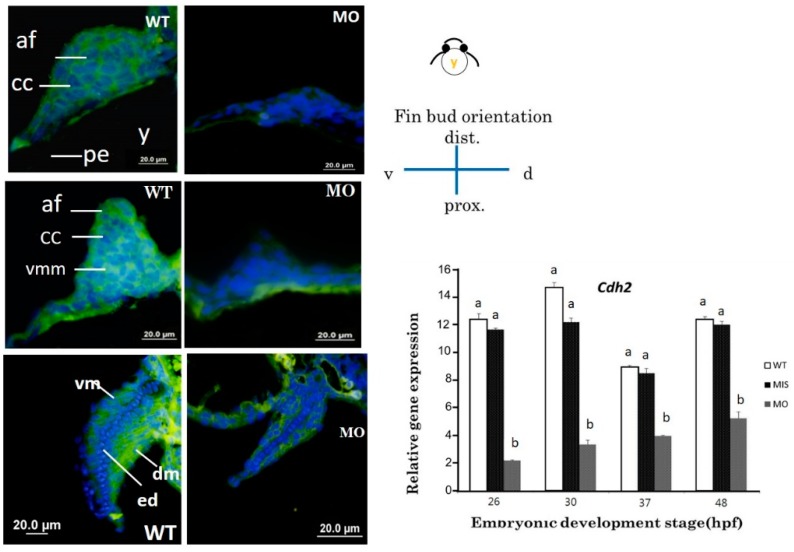
Cross sections of pectoral fins subjected to immunofluorescence staining for CDH2. Pectoral fins of wild-type embryos showed abundant N-cadherin surrounding chondrocytes at 37 (**A**) and 48 hpf (**C**). N-cadherin is observable in the dorsal and ventral muscular region surrounding the endoskeletal disc at 72 hpf (**E**). The *tbx5a* morphant with a stubby pectoral fin showed a lack of N-cadherin at 37 hpf (**B**), 48 hpf (**D**) and 72 hpf (**F**). *cdh2* expression was significantly inhibited in the tbx5 morphant group at 26, 30, 37, and 48 hpf (**G**) (*n* = 50 embryos, triplet). (cl, cleithrum; cc, chondrogenic condensation; dm, dorsal musculature; ed, endoskeleton disc; sco, scapulocoracoid; vm, ventral musculature; blue, nuclear; green, CDH2 protein, WT: Uninjected group, MIS: Control MO-injected group, MO: tbx5 morphant group, a, b: A significant difference was detected by one-way ANOVA with Duncan’s multiple range test.

**Figure 6 jdb-07-00015-f006:**
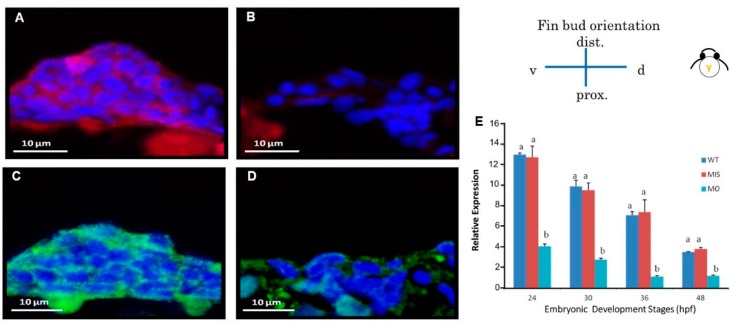
The immunochemical characterization of fibronectin and HA-binding protein in fin tissue at 48 hpf. Compared with wild-type embryos (**A**,**C**), *tbx5a* MO embryos showed decreased expression of fibronectin and HA-binding protein at 48 hpf (**B**,**D**). *Cx43* expression in the *tbx5a* morphant group was significantly inhibited at 24, 30, 36, and 48 hpf (*n* = 50, triplet) (**E**). a, b: A significant difference was detected by One-way ANOVA with Duncan’s multiple range test; red in (**A**,**B**), fibronectin; green in (**C**,**D**), HA-binding protein; blue: nuclear, WT: Uninjected group, MIS: Control MO-injected group, MO: *tbx5a* morphant group.

**Figure 7 jdb-07-00015-f007:**
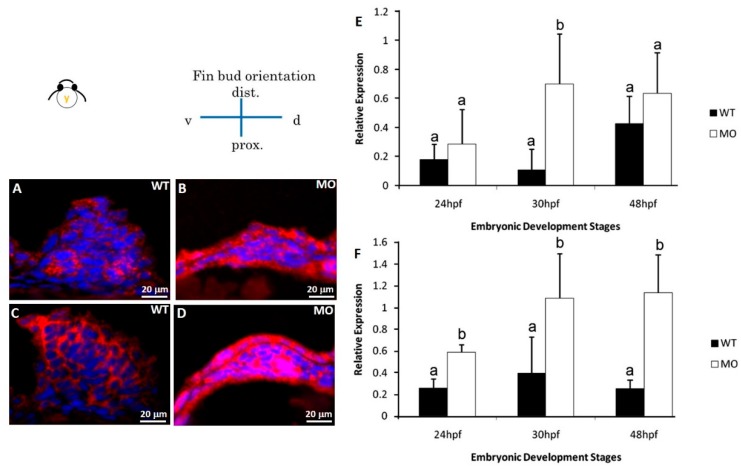
Immunohistochemical staining of apoptosis-related proteins. Pectoral fins were stained with apoptosis-related (BAD and BCL2) antibodies (red) and counterstained with DAPI (blue) to observe the nucleus. BAD (**B**) and BCL2 (**D**) proteins were significantly uninhibited in the *tbx5a* morphant group compared with those in the Uninjected group (**A**,**C**). The embryo anterior is shown on the left. The *Bad* (**E**) and *Bcl2* (**F**) expression in the *tbx5a* morphant group was significantly uninhibited at 24, 30, 36, and 48 hpf (*n* = 50, triplet) (**E**). hpf, hours postfertilization; WT, Uninjected group; MO, *tbx5a* morphant group. a, b: A significant difference was detected by a, b: A significant difference was detected by One-way ANOVA with Duncan’s multiple range test.

**Table 1 jdb-07-00015-t001:** Cartilage and fin development-related gene transcripts downregulated 1.5x in tbx5 knockdown embryos in different embryonic developmental stages.

Genbank Accession	Gene Symbol	Gene Name	Biological Process	Log2 Expression Ratio	Stages
NM_131038	*cx43*	connexin 43	fin development	−1.98	24
NM_001080012	*hhip*	hedgehog interacting protein	fin development	−1.63	24
NM_001077145	*bbs7*	Bardet-Biedl syndrome 7	fin development	−2.02	24
NM_194416	*mmp14a*	matrix metalloproteinase 14 (membrane-inserted) alpha	cartilage development	−1.67	24
NM_199777	*sec23b*	Sec23 homolog B (S. cerevisiae)	cartilage development	−1.88	24
NM_176859	*tfap2a*	transcription factor AP-2 alpha	cartilage development	−2.12	24
NM_131503	*axin 1*	axin 1	fin development	−1.71	30
NM_131260	*msxb*	muscle segment homeobox B	fin development	−1.61	30
NM_001005771	*hoxc8a*	homeo box C8a	fin development	−1.93	30
NM_205635	*slc35b2*	solute carrier family 35, member B2	cartilage development	−1.89	30
NM_131306	*dlx5a*	distal-less homeobox gene 5a	cartilage development	−1.52	30
NM_131305	*dlx1a*	distal-less homeobox gene 1a	cartilage development	−1.84	30
NM_001024665	*tfap2b*	transcription factor AP-2 beta	cartilage development	−1.77	30
NM_152963	*fmr1*	fragile X mental retardation 1	cartilage development	−1.53	30
BC163560	*runx3*	runt-related transcription factor 3	cartilage development	−1.65	30
NM_131081	*cdh2*	cadherin 2, neuronal	cartilage condensation	−1.94	30
NM_001126448	*lect1*	leukocyte cell derived chemotaxin 1	cartilage development	−1.75	30
NM_001110278	*acvr2a*	activin receptor IIa	cartilage development	−1.99	30
NM_194414	*mmp14b*	matrix metalloproteinase 14 (membrane-inserted) beta	cartilage development	−1.87	30

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
