# Peer review of "Pectoral Fin Anomalies in tbx5a Knockdown Zebrafish Embryos Related to the Cascade Effect of N-Cadherin and Extracellular Matrix Formation"

_jdb, 2019, doi:10.3390/jdb7030015_

Round 1

Reviewer 1 Report

The revised manuscript successfully addresses many of my previous concerns.  The only remaining significant problem is that it is a simple morpholino knockdown experiment without any rescue or ectopic over-expression data to validate the interpretation.  Ectopically expressing Tbx5a in lateral plate mesoderm outside of the limb bud and inducing some of the genes you see down regulated in your morphants would make this a far more compelling study.  However, apart from that, my remaining criticisms are all relatively minor.

1) Details of image process are still missing from the methods section

2) There seems to be something wrong with the scale bars in fig 6 and/or 7; images appear to be very close to the same scale (based on number and size of nuclei) but fig 6 has 10 µm scale bars that are substantially bigger than the 50 µm scale bars in fig 7. And there is no need for third significant digit in scale bars of fig 5.

3) Throughout the manuscript, the control group should not be referred to as the "WT group", as all the embryos in this study are "wildtype".  The embryos should be described as "Uninjected", "Control MO injected", or "tbx5a MO injected" (or "tbx5a morphant").

4) Lines 193 and 195: “ventral view” should be “dorsal view”

5) Line 292: “immunohistochemical” should be “immunofluorescent”

6) Line 204: “…total number of screening genes was 43663” - there are only about 26,000 protein coding genes in the zebrafish genome… presumably this should say something like “…43,663 probe sequences, representing X genes.”  Similarly, in the rest of the section regarding the microarray results, the number of “genes” up or down in the tbx5a knockdowns needs to be changed to “sequences”.  Which brings up the question, were any of the up or down regulated genes identified by multiple targets, and if so, was the up/down data consistent? Is there any indication of changes in splicing (e.g. one of a set of probes for a given gene going up or down less than another probe for the same gene)?

Finally, WRT the microarray data, please include the relative expression values for the genes listed in table 1.

7) Line 356: “One of the earliest events in chondrogenesis in the pre-CC stage represents the aggregation of chondro-progenitor mesenchymal cells.” should read “One of the earliest events in chondrogenesis in the pre-CC stage is the aggregation of chondro-progenitor mesenchymal cells.”

8) Line 363: “evokes” should be “provokes expression of”

9) Line 382: “…the tbx5 gene has a direct or/and indirect influence on cdh2 gene expression…” should be “…expression of tbx5a affects expression of cdh2…”

10) Line 396: delete “Nevertheless”

11) Line 405: “The pectoral fin disk cartilage of zebrafish is unique because of its strong fibronectin-positive ECM” should read “The pectoral fin disk cartilage of zebrafish is notable for its fibronectin-rich ECM”

12) Line 357: “The microarray screening results in this report provided strong evidence indicating that all genes related to fin development were downregulated in the MO group at the pre-CC stage (24 hpf) and early CC stage (30 hpf).”  Firstly, this is a significant overstatement of the results; we don’t even know what all the genes involved in fin development are, and your data only shows downregulation of a dozen.  Secondly, this is a great example of the terrible language that results from the use of past-tense passive voice grammatical constructions.  I suggest “Several genes involved in chondrogenesis are downregulated in tbx5a morphants at 24 and 30 hpf.”  Similarly, on line 422: “all fin development-related genes” should be “several fin development-related genes”

And finally, in their response to my previous review, the authors wrote “The previous manuscript was edited by American Journal Experts and the newest manuscript was re-edited by this Co.”

I understand that English is not the authors’ native language and genuinely appreciate the effort the authors have gone to in order to achieve an acceptable standard of English language use in their drafts; the current version is significantly improved in this regard.  However, there remain a significant number of issues with the writing that should be brought to the attention of the editing company.  For example, in the introduction: 

Line 43 “highly preserved” should read “highly conserved”

Line 57 - genes don’t induce things; expression of gene products may be [necessary] parts of the processes that we observe during development.  So rather than “The tbx5a gene induces mesenchymal cell migration…” we should say “The mesodermal mesenchymal cell migration during fin formation, which plays crucial role in the localization and timing of induction of the pectoral fin bud, depends on expression of the tbx5a gene.”

Line 59 - similarly, genes don’t conduct signals.  And given that the tbx5a gene encodes a transcription factor, it’s not even reasonable to say that the Tbx5a protein conducts signals between the AER and the underlying mesoderm.  Presumably Tbx5a promotes the expression of signalling molecules, their receptors and the associated signal transduction machinery necessary for these signals to be generated and received, but the Tbx5a protein isn’t doing it, much less the tbx5a gene.

Avoid superfluous articles ('the', 'a', 'an') and use active voice and present tense as much as possible. For example, on Line 60 rather than “The tbx5a expression was maintained at a low level up to 96 hpf” say “Tbx5a expression is maintained at a low level until 96 hpf.”  Similarly, throughout the manuscript, avoid phrases like “could be observed” (e.g. line 249… just say “are visible” or something).  Also, the Introduction should end with a thesis-statement or some type of summary of the findings.  So instead of “Cartilage-related gene expression in tbx5a gene knockdown zebrafish embryos was studied by DNA microarrays and immuno-histology…” say something like “We use microarrays, RTqPCR, immunofluorescence and histology to show that expression of several genes associated with chondrogenesis are downregulated in tbx5a morphant embryos at the transcriptional and/or protein levels. We conclude that tbx5a is essential for the molecular mechanisms underlying CC in pectoral fin development.”

All of these are minor, stylistic/editorial issues, but they add up to a much more readable and understandable manuscript, and should be corrected by any competent scientific editor.  I suggest you express your dissatisfaction with American Journal Experts (feel free to provide them with this commentary) and find another company for your future work.

Author Response

s attached file

Reviewer 2 Report

This revised manuscript has adequately addressed the concerns/issues raised in the review of the original manuscript.  There are 2 new, yet minor, issues that should be addressed.

1) Legend for Figure 7 - Bad (E) and Bcl2 (F) expression in tbx5a MO group was significantly increased.... not inhibited as a stated. 

2) Results of statistical analyses as shown in figure legends are not clear.  Specifically which groups are different from one another?  What is the meaning of symbols a or b (figures 1,6 and 7) or * or ** or # (figure 5)?  Since one-way ANOVA was used, it should be included in statistical analysis in the Materials and Methods section.  Furthermore, what pos-hoc test(s) was/were used to assess differences between specific groups?

Author Response

s attached file

This manuscript is a resubmission of an earlier submission. The following is a list of the peer review reports and author responses from that submission.

Round 1

Reviewer 1 Report

The manuscript by Lu et al. presents some potentially interesting, but largely descriptive data documenting changes in expression of a number of genes involved in fin/limb-bud development after morpholino-mediated knock-down of Tbx5a in zebrafish.  There are four major issues with the present submission, as well as a large number of minor issues.

1) Unfortunately, as is often the case with morpholino-based studies, the author’s own data show dramatic and pervasive off-target effects (e.g. extensive apoptosis throughout the embryo), making these results difficult to interpret.  The authors do a good job of demonstrating that their morpholino does target the Tbx5a protein using western blots, and the surviving morphant embryos appear morphologically normal (although only anterior views were shown), suggesting there is a specific effect mixed in here.  However, there is clearly a non-specific effect as both the Tbx5a and mismatch control morpholino-injected embryos had poor (<50%) survival, and the pervasive apoptosis illustrated in figures 6 and 7 leave the reader with no choice but to assume there are confounding effects in all of these data.  Given the efficacy and low cost of CRISPR based mutagenesis, have to wonder why the authors did not make a tbx5a mutant with which to pursue these investigations.

2) More critically, some of the data is not of adequate quality for publication.  Specifically, the cadherin immunofluorescence micrographs depict staining that is very diffuse and looks non-specific.  Cadherin staining should clearly outline cells (adherens junctions). The signal looks strongest in yolk, which is almost certainly an artifact.  As an important aside to this point, there is no information in the Methods section pertaining to the microscopy and image processing - what type of microscope (and more importantly what objective lens and detector systems) were used, what image processing was done (using what software), etc.  Immunofluorescence is notoriously difficult to use in making quantitative comparisons - it’s easy to say that the distribution of an antigen is different between two samples, but it much more difficult to say that the abundance of that antigen is significantly different on the basis of fluorescent intensity.

3) The major conclusion the authors draw “Tbx5a deficiency induced a downregulation of all fin development-related genes, which interfered with mesenchymal cell migration and proliferation leading to CC in pectoral fin buds” is likely incorrect. Tbx5a knockdown likely results in reduced expression of FGF8 by the AER, which in turn results in reduced proliferation and migration of mesoderm, resulting in reduced expression of genes related to fin development.  The authors need to address wether the loss of Tbx5a results in loss of AER signalling to the underlying mesoderm, and wether the reduced mesodermal component of the fin buds is due to reduced proliferation (e.g. by BrdU labeling), increased apoptosis, and/or decreased immigration of mesenchyme from the lateral plate.

Similarly, the authors state “We report herein, for the first time, that the tbx5 gene has a direct influence on cdh2 gene expression” No data to this effect is presented; cdh2 expression is down in the Tbx5a morphants, but that could be (and I expect is) due to indirect effects.

4) I would normally make this a minor issue, but the number of English language, formatting, and basic mechanical issues (changing fonts, missing or extra spaces, missing super/subscripts, etc.) with this manuscript are so numerous and extensive that it made it difficult to understand.  For example in figure 1 the graph showing the “Non-fin defect rate” initially confused me, as I interpreted it to be showing the number of embryos in which defects were observed in tissues outside the fins.  But given that 100% of the untreated embryos apparently had “non-fin defects” that made no sense.  I assume this graph is meant to represent the number of embryos in each treatment group that have normal fins.  I would insist that this manuscript undergo english language editing before re-reviewing it.

Minor issues (in no particular order):

1) Unc45a is a myosin chaperone, not a cartilage development gene.

2) Figure 1K Cdh2 expression - indicate that this is RTqPCR in figure legend and add reference to panel K in legend.

3) Why different colours and different fonts in panels J and K?

4) No scale bars on Fig 1 f, g, h, i

5) labels and scale bars often distorted due to adjustments of image aspect ratio/size; this can be misleading.  Elaborate on all image processing in methods section.

Reviewer 2 Report

See attached file
